# Learning Stable Classifiers by Transferring Unstable Features

## Abstract

While unbiased machine learning models are essential for many applications, bias is a human-defined concept that can vary across tasks. Given only input-label pairs, algorithms may lack sufficient information to distinguish stable (causal) features from unstable (spurious) features. However, related tasks often share similar biases – an observation we may leverage to develop stable classifiers in the transfer setting. In this work, we explicitly inform the target classifier about unstable features in the source tasks. Specifically, we derive a representation that encodes the unstable features by contrasting different data environments in the source task. We achieve robustness by clustering data of the target task according to this representation and minimizing the worst-case risk across these clusters. We evaluate our method on both text and image classifications. Empirical results demonstrate that our algorithm is able to maintain robustness on the target task for both synthetically generated enviornments and real-world environments. Our code will be publicly available.

## 1 Introduction

Automatic de-biasing (Sohoni et al., 2020; Creager et al., 2021; Sanh et al., 2021) has emerged as a promising direction for learning stable classifiers. The key premise here is that no additional annotations for the bias attribute are required. However, bias is a human-defined concept and can vary from task to task. Provided with only input-label pairs, algorithms may not have sufficient information to distinguish stable (causal) features from unstable (spurious) features.

To address this challenge, we note that related tasks are often fraught with similar spurious correlations. For instance, when classifying animals such as camels vs. cows, their backgrounds (desert vs. grass) may constitute a spurious correlation (Beery et al., 2018). The same bias between the label and the background also persists in other related classification tasks (such as sheep vs. antelope). In the resource-scarce target task, we only have access to the input-label pairs. However, in the source tasks, where training data is sufficient, identifying biases may be easier. For instance, we may have examples collected from multiple environments, in which correlations between bias features and the label are different (Arjovsky et al., 2019). These source environments help us define the exact bias features that we want to regulate.

One obvious approach to utilize the source task is direct transfer. Specifically, given multiple source environments, we can train an unbiased source classifier and then apply its representation to the target task. However, we empirically demonstrate that while the source classifier is not biased when making its final predictions, its internal continuous representation can still encode information about the unstable features. Figure 1 shows that in Colored MNIST, where the digit label is spuriously correlated with the image color, direct transfer by either re-using or fine-tuning the representation learned on the source task fails in the target task, performing no better than the majority baseline.

In this paper, we propose to explicitly inform the target classifier about unstable features from the source data. Specifically, we derive a representation that encodes these unstable features using the source environments. Then we identify distinct subpopulations by clustering examples based on this representation and apply group DRO (Sagawa et al., 2019) to minimize the worst-case risk over these subpopulations. As a result, we enforce the target classifier to be robust against different values of the unstable features. In the example above, animals would be clustered according to backgrounds, and the classifier should perform well regardless of the clusters (backgrounds).

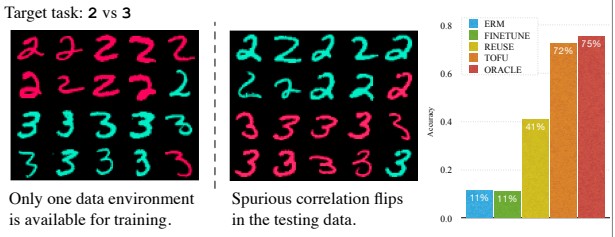

Figure 1: Transferring across tasks in Colored MNIST (Arjovsky et al., 2019). On the source task, we learn a color-invariant model that achieves oracle performance (given direct access to the unstable features). However, directly transferring this model to the target task, by reusing or fine-tuning its feature extractor, severely overfits the spurious correlation and underperforms the majority baseline (50%) on a test set where the spurious correlation flips. By explicitly transferring the unstable features, our algorithm TOFU (Transfer OF Unstable features) is able to reach the oracle performance.

The remaining question is how to compute the unstable feature representation using the source data environments. Following Bao et al. (2021), we hypothesize that unstable features are reflected in mistakes observed during classifier transfer across environments. For instance, if the classifier uses the background to distinguish camels from cows, the camel images that are predicted correctly would have a desert background while those predicted incorrectly are likely to have a grass background. More generally, we prove that among examples with the same label value, those with the same prediction outcome will have more similar unstable features than those with different predictions. By forcing examples with the same prediction outcome to stay closer in the feature space, we obtain a representation that encodes these latent unstable features.

We evaluate our approach, Transfer OF Unstable features (TOFU), on both synthetic and real-world environments. Our synthetic experiments first confirm our hypothesis that standard transfer approaches fail to learn a stable classifier for the target task. By explicitly transferring the unstable features, our method significantly improves over the best baseline across 12 transfer settings (22.9% in accuracy), and reaches the performance of an oracle model that has direct access to the unstable features (0.3% gap). Next, we consider a practical setting where environments are defined by an input attribute and our goal is to reduce biases from other unknown attributes. On CelebA, TOFU achieves the best worst-group accuracy across 38 latent attributes, outperforming the best baseline by 18.06%. Qualitative and quantitative analyses confirm that TOFU is able to identify the unstable features.

## 2 RELATED WORK

**Removing bias via annotations:**  Due to idiosyncrasies of the data collection process, annotations are often coupled with unwanted biases (Buolamwini and Gebru, 2018; Schuster et al., 2019; McCoy et al., 2019; Yang et al., 2019). To address this issue and learn robust models, researchers leverage extra information (Belinkov et al., 2019; Stacey et al., 2020; Hinton, 2002; Clark et al., 2019; He et al., 2019; Mahabadi et al., 2020). One line of work assumes that the bias attributes are known and have been annotated for each example, e.g., group distributionally robust optimization (DRO) (Hu et al., 2018; Oren et al., 2019; Sagawa et al., 2020). By defining groups based on these bias attributes, we explicitly specify the distribution family to optimize over. However, identifying the hidden biases is time-consuming and often requires domain knowledge (Zellers et al., 2019; Sakaguchi et al., 2020). To address this issue, another line of work (Peters et al., 2016; Krueger et al., 2020; Chang et al., 2020; Jin et al., 2020; Ahuja et al., 2020; Arjovsky et al., 2019; Bao et al., 2021; Kuang et al., 2020; Shen et al., 2020) only assumes access to a set of data environments. These environments are defined based on readily-available information of the data collection circumstances, such as location and time. The main assumption is that while spurious correlations vary across different environments, the association between the causal features and the label should stay the same. Thus, by learning a representation that is invariant across all environments, they alleviate the dependency on spurious features. In contrast to previous works, we don't have access to any additional information besides the labels in our target task. We show that we can achieve robustness by transferring the unstable features from a related source task.

**Automatic de-biasing**    A number of recent approaches focus on a more common setting where the algorithm only has access to the input-label pairs. Sanh et al. (2021); Nam et al. (2020); Utama et al. (2020) find that weak models are more vulnerable to spurious correlations as they only learn shallow heuristics. By boosting from their mistakes, they obtain a more robust model. Qiao et al. (2020) uses adversarial learning to augment the biased training data. Creager et al. (2021); Sohoni et al. (2020); Ahmed et al. (2020); Matsuura and Harada (2020); Liu et al. (2021) propose to identify minority groups by looking at the features produced by a biased model.

These automatic approaches are intriguing as they do not require additional annotation. However, we note that bias is a human-centric concept and can vary from tasks to tasks. For models that only have access to the input-label pairs, they have no information to distinguish causal features from bias features. For example, consider the Colored MNIST dataset, where color and digit shape are correlated in the training set but not in the test set. If our task is to predict the digit, then color becomes the spurious bias that we want to remove. Vice versa, if we want to predict the color, then digit shape is spurious. Creager et al. (2021); Nam et al. (2020) empirically demonstrate that they can learn a color-invariant model for the digit prediction task. However, their approaches will result in the same color-invariant model for the color prediction task, and thus fail at test time, when color and digit are no longer correlated. In this work, we leverage source tasks to define the exact bias that we want to remove for the target task.

**Transferring robustness across tasks:**    Prior work has also studied the transferability of adversarial robustness across tasks. For example, Hendrycks et al. (2019); Shafahi et al. (2020) show that by pre-training the model on a large-scale source task, we can improve the model robustness against adversarial perturbations over $l_\infty$ norm. We note that these perturbations measure the smoothness of the classifier, rather than the stability of the classifier against spurious correlations. In fact, our results show that if we directly re-use or fine-tune the pre-trained feature extractor on the target task, the model will quickly over-fit to the unstable correlations present in the data. We propose to address this issue by explicitly inferring the unstable features using the source environments and use this information to guide the target classifier during training.

## 3    METHOD

**Problem formulation**    We consider the transfer problem from a source task to a target task. For the source task, we assume the standard setting (Arjovsky et al., 2019) where the training data contain $n$ environments $E_1, \ldots, E_n$. Within each environment $E_i$, examples are drawn from the joint distribution $P_i(x, y)$. Following Woodward (2005), we define unstable features $\mathcal{Z}(x)$ as features that are *differentially* correlated with the label across the environments. We note that $\mathcal{Z}(x)$ is unknown to the model.

For the target task, we only have access to the input-label pairs $(x, y)$ (i.e. no environments). We assume that the target label is *not* causally associated with the above unstable features $\mathcal{Z}$. However, due to collection biases, the target data may contain *spurious correlations* between the label and $\mathcal{Z}$. Our goal is to transfer the knowledge that $\mathcal{Z}$ is unstable in the source task, so that the target classifier will not rely on these spurious features.

**Overview**    If the unstable features have been identified for the target task, we can simply apply group DRO to learn a stable classifier. By grouping examples based on the unstable features and minimizing the worst-case risk over these *manually-defined* groups, we explicitly address the bias from these unstable features (Hu et al., 2018; Oren et al., 2019; Sagawa et al., 2020). In our setup, while these unstable features are not accessible, we can leverage the source environments to derive groups over the target data that are informative of these biases. Applying group DRO on these *automatically-derived* groups, we can eliminate the unstable correlations in the target task.

Our overall transfer paradigm is depicted in Figure 2. It consists of two steps: inferring unstable features from the source task (Section 3.1) and learning stable correlations for the target task (Section 3.2). First, for the source task we use a classifier trained on one environment to partition data from another environment based on the correctness of its predictions. Starting from the theoretical results in (Bao et al., 2021), we show that these partitions reflect the similarity of the examples in terms of their unstable features: among examples with the same label value, those that share the same prediction

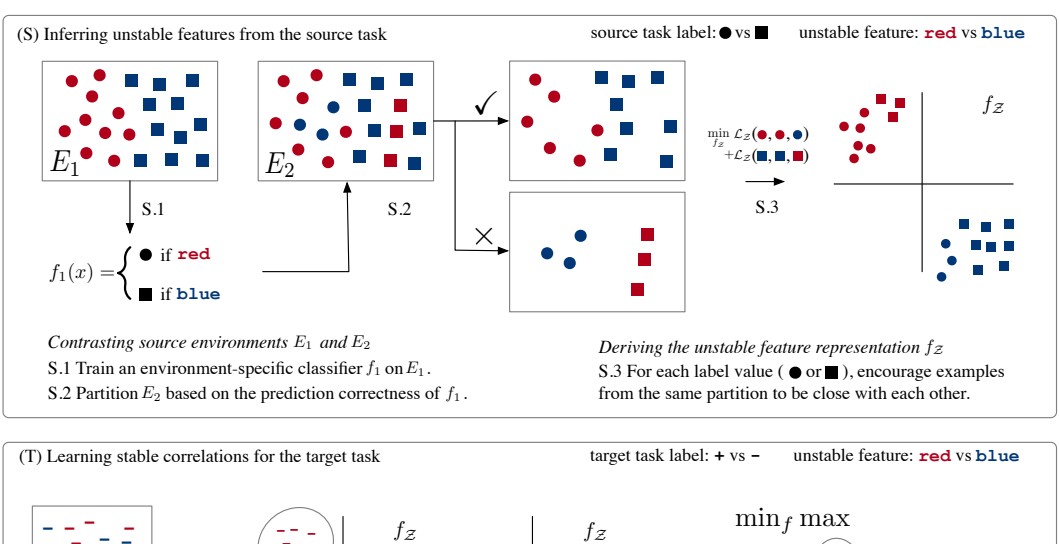

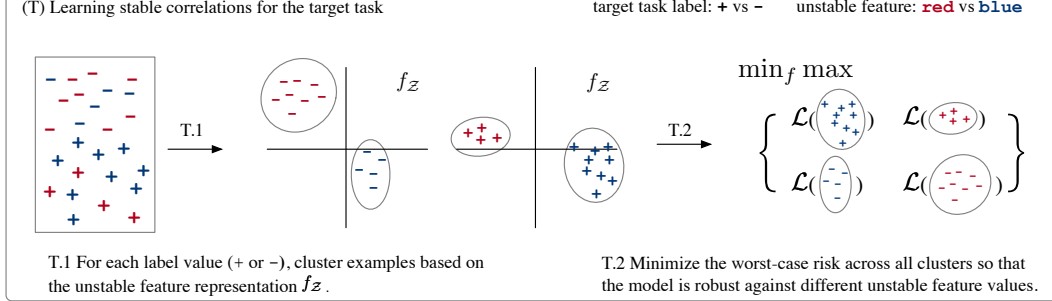

Figure 2: Our algorithm TOFU 1) infers unstable features from the source task (Section 3.1) and 2) learns stable correlations for the target task (Section 3.2). We create partitions for all environment pairs. For ease of illustration, we only depict using $f_1$ to partition $E_2$. Best viewed in color.

outcome have more similar unstable features than those with different predictions (Theorem 1). We can then derive a representation $f_{\mathcal{Z}}$ where examples are distributed based on the unstable features $\mathcal{Z}$. Next, we cluster target examples into groups based on the learned unstable feature representation $f_{\mathcal{Z}}$. These *automatically-derived* groups correspond to different modes of the unstable features, and they act as proxies to the *manually-defined* groups in the oracle setting where unstable features are explicitly annotated. Finally, we use group DRO to obtain our robust target classifier by minimizing the worst-case risk over these groups.

## 3.1 INFERRING UNSTABLE FEATURES FROM THE SOURCE TASK

Given the data environments from the source task, we would like to 1) identify the unstable correlations across these environments; 2) learn a representation $f_{\mathcal{Z}}(x)$ that encodes the unstable features $\mathcal{Z}(x)$. We achieve the first goal by contrasting the empirical distribution of different environments (Figure 2.S.1 and Figure 2.S.2) and the second goal by metric learning (Figure 2.S.3).

Let $E_i$ and $E_j$ be two different data environments. Bao et al. (2021) shows that by training a classifier $f_i$ on $E_i$ and using it to make predictions on $E_j$, we can reveal the unstable correlations from its prediction results. Intuitively, if the unstable correlations are stronger in $E_i$, the classifier $f_i$ will overuse these correlations and make mistakes on $E_j$ when these stronger correlations do not hold.[1]

In this work, we connect the prediction results directly to the unstable features. We show that the prediction results of the classifier $f_i$ on $E_j$ estimate the relative distance of the unstable features.

**Theorem 1** (Simplified). *Consider examples in $E_j$ with label value $y$. Let $X_1^{\checkmark}, X_2^{\checkmark}$ denote two batches of examples that $f_i$ predicted correctly, and let $X_3^{\times}$ denote a batch of incorrect predictions. We use $\overline{\cdot}$ to represent the mean across a given batch. Following the same assumption in (Bao et al., 2021), we have*

$$\|\overline{\mathcal{Z}}(X_1^{\checkmark}) - \overline{\mathcal{Z}}(X_2^{\checkmark})\|_2 < \|\overline{\mathcal{Z}}(X_1^{\checkmark}) - \overline{\mathcal{Z}}(X_3^{\times})\|_2$$

---

[1]We assume that the distributions of $E_i$ and $E_j$ are different enough such that $f_i$ will make mistakes.

*almost surely for large enough batch size.*[2].

The result makes intuitive sense as we would expect example pairs that share the same prediction outcome should be more similar than those with different prediction outcomes. We note that it is critical to look at examples with the same label value; otherwise, the unstable features will be coupled with the task-specific label in the prediction results.

While the value of the unstable features $\mathcal{Z}(x)$ is still not directly accessible, Theorem 1 enables us to learn a feature representation $f_{\mathcal{Z}}(x)$ that preserves the distance between the examples in terms of their unstable features. We adopt standard metric learning (Chechik et al., 2010) to minimize the following triplet loss:

$$\mathcal{L}_{\mathcal{Z}}(X_1^{\checkmark}, X_2^{\checkmark}, X_3^{\times}) = \max(0, \|\overline{f_{\mathcal{Z}}}(X_1^{\checkmark}) - \overline{f_{\mathcal{Z}}}(X_2^{\checkmark})\|_2^2 - \|\overline{f_{\mathcal{Z}}}(X_1^{\checkmark}) - \overline{f_{\mathcal{Z}}}(X_3^{\times})\|_2^2 + \delta), \quad (1)$$

where $\delta$ is a hyper-parameter. By minimizing Eq equation 1, we encourage examples that have similar unstable features to be close in the representation $f_{\mathcal{Z}}$. To summarize, inferring unstable features from the source task consists of three steps (Figure 2.S):

**S.1** For each source environment $E_i$, train an environment-specific classifier $f_i$.

**S.2** For each pair of environments $E_i$ and $E_j$, use classifier $f_i$ to partition $E_j$ into two sets: $E_j^{i\checkmark}$ and $E_j^{i\times}$, where $E_j^{i\checkmark}$ contains examples that $f_i$ predicted correctly and $E_j^{i\times}$ contains those predicted incorrectly.

**S.3** Learn an unstable feature representation $f_{\mathcal{Z}}$ by minimizing Eq equation 1 across all pairs of environments $E_i, E_j$ and all possible label value $y$:

$$f_{\mathcal{Z}} = \arg\min \sum_{y, E_i \neq E_j} \mathbb{E}_{X_1^{\checkmark}, X_2^{\checkmark}, X_3^{\times}} \left[ \mathcal{L}_{\mathcal{Z}}(X_1^{\checkmark}, X_2^{\checkmark}, X_3^{\times}) \right],$$

where batches $X_1^{\checkmark}, X_2^{\checkmark}$ are sampled uniformly from $E_j^{i\checkmark}|_y$ and batch $X_3^{\times}$ is sampled uniformly from $E_j^{i\times}|_y$ ($\cdot|_y$ denotes the subset of $\cdot$ with label value $y$).

## 3.2 Learning stable correlations for the target task

Given the unstable feature representation $f_{\mathcal{Z}}$, our goal is to learn a target classifier that focuses on the stable correlations rather than using unstable features. Inspired by group DRO (Sagawa et al., 2020) we minimize the worst-case risk across groups of examples that are representative of different unstable feature values. However, in contrast to DRO, these groups are constructed automatically based on the previously learned representation $f_{\mathcal{Z}}$.

For each target label value $y$, we use the representation $f_{\mathcal{Z}}$ to cluster target examples with label $y$ into different clusters (Figure 2.T.1). Since these clusters capture different modes of the unstable features, they are approximations of the typical manually-defined groups when annotations of the unstable features are available. By minimizing the worst-case risk across all clusters, we explicitly enforce the classifier to be robust against unstable correlations (Figure 2.T.2). We note that it is important to cluster within examples of the same label, as opposed to clustering the whole dataset. Otherwise, the cluster assignment may be correlated with the target label.

Concretely, learning stable correlations for the target task has two steps (Figure 2.T).

**T.1** For each label value $y$, apply K-means ($l_2$ distance) to cluster examples with label $y$ in the feature space $f_{\mathcal{Z}}$. We use $C_1^y, \ldots, C_{n_c}^y$ to denote the resulting cluster assignment, where $n_c$ is a hyper-parameter.

**T.2** Train the target classifier $f$ by minimizing the worst-case risk over all clusters:

$$f = \arg\min \max_{i,y} \mathcal{L}(C_i^y),$$

where $\mathcal{L}(C_i^y)$ is the empirical risk on cluster $C_i^y$.

Table 1: Pearson correlation coefficient between the spurious feature $\mathcal{Z}$ and the label $Y$ for each task. The validation environment $E^{\text{val}}$ follows the same distribution as $E_1^{\text{train}}$. We study the transfer problem between different task pairs. For the source task $S$, the model can access $E_1^{\text{train}}(S), E_2^{\text{train}}(S)$ and $E^{\text{val}}(S)$. For the target task $T$, the model can access $E_1^{\text{train}}(T)$ and $E^{\text{val}}(T)$.

| $\rho(\mathcal{Z}, Y)$ | MNIST | | BEER REVIEW | | | ASK2ME | | WATERBIRD | |
|---|---|---|---|---|---|---|---|---|---|
| | ODD | EVEN | LOOK | AROMA | PALATE | PENE. | INCI. | WATER | SEA. |
| $E_1^{\text{train}}$ | 0.87 | 0.87 | 0.60 | 0.60 | 0.60 | 0.31 | 0.44 | 0.36 | 0.39 |
| $E_2^{\text{train}}$ | 0.75 | 0.75 | 0.80 | 0.80 | 0.80 | 0.52 | 0.66 | 0.63 | 0.64 |
| $E^{\text{val}}$ | 0.87 | 0.87 | 0.60 | 0.60 | 0.60 | 0.31 | 0.44 | 0.36 | 0.39 |
| $E^{\text{test}}$ | $-0.11$ | $-0.11$ | $-0.80$ | $-0.80$ | $-0.80$ | 0.00 | 0.00 | 0.00 | 0.00 |

## 4 EXPERIMENTAL SETUP

### 4.1 DATASETS AND SETTINGS

**Synthetic environments** We start with controlled experiments where environments are created based on the spurious correlation. We consider four datasets: MNIST (LeCun et al., 1998), Beer-Review (McAuley et al., 2012), ASK2ME (Bao et al., 2019a) and Waterbird (Sagawa et al., 2019). In MNIST and BeerReview, we inject spurious feature to the input (background color for MNIST and pseudo token for BeerReview). In ASK2ME and Waterbird, spurious feature corresponds to an attribute of the input (`breast_cancer` for ASK2ME and `background` for Waterbird).

For each dataset, we consider multiple tasks and study the transfer between these tasks. Specifically, for each task, we split its data into four environments: $E_1^{\text{train}}, E_2^{\text{train}}, E^{\text{val}}, E^{\text{test}}$, where spurious correlations vary across the two training environments $E_1^{\text{train}}, E_2^{\text{train}}$. For the source task $S$, the model can access both of its training environments $E_1^{\text{train}}(S), E_2^{\text{train}}(S)$. For the target task $T$, the model only has access to one training environment $E_1^{\text{train}}(T)$. We note that the validation set $E^{\text{val}}(T)$ plays an important role in early-stopping and hyper-parameter tuning, especially when the distribution of the data is different between training and testing (Gulrajani and Lopez-Paz, 2020). In this work, since we don't have access to multiple training environments on the target task, we assume that the validation data $E^{\text{val}}$ follows the same distribution as the training data $E_1^{\text{train}}$. Table 1 summarizes the level of the spurious correlations for different tasks. Additional details can be found in Appendix C.1.[3]

**Natural environments** We also consider a practical setting where environments are directly defined by a given attribute of the input, and our goal is to reduce model biases from other latent attributes. We study CelebA (Liu et al., 2015a) where each input (an image of a human face) is annotated with 40 binary attributes. The source task is to predict the `Eyeglasses` attribute and the target task is to predict the `BlondHair` attribute. We use the `Young` attribute to define two environments: $E_1 = \{\text{Young} = 0\}$ and $E_2 = \{\text{Young} = 1\}$. In the source task, both environments are available. In the target task, we only have access to environment $E_1$ during training and validation. At test time, we evaluate the robustness of our target classifier against other latent attributes. Specifically, for each unknown attribute such as `Male`, we partition the testing data into four groups: $\{\text{Male} = 1, \text{BlondHair} = 0\}, \{\text{Male} = 0, \text{BlondHair} = 0\}, \{\text{Male} = 1, \text{BlondHair} = 1\}, \{\text{Male} = 0, \text{BlondHair} = 1\}$. Following Sagawa et al. (2019), We report the worst-group accuracy and the average-group accuracy.

### 4.2 BASELINES

We compare our algorithm against the following baselines. For fair comparison, all methods share the same representation backbone and hyper-parameter search space.

**ERM baseline** We learn a classifier on the target task from scratch by minimizing the average loss across all examples. Note that this classifier is independent of the source task. Its performance reflects the deviation between the training distribution and the testing distribution of the target task.

---

[2]See Appendix B for the full theorem and proof.

[3]All data splits, hyper-parameter search spaces are available in the supplementary materials.

Table 2: Target task accuracy of different methods. All methods are tuned based on a held-out validation set that follows from the same distribution as the target training data. Bottom right: standard deviation across 5 runs. Upper right: source task testing performance (if applicable).

| | SOURCE | TARGET | ERM | REUSE$_{\text{PI}}$ | FINETUNE$_{\text{PI}}$ | MULTITASK | TOFU | ORACLE |
|---|---|---|---|---|---|---|---|---|
| MNIST | ODD | EVEN | $12.3_{\pm0.6}$ | $14.4^{(70.9)}_{\pm1.0}$ | $11.2^{(70.1)}_{\pm2.1}$ | $11.6^{(69.6)}_{\pm0.6}$ | $\mathbf{69.1}_{\pm1.6}$ | $68.7_{\pm0.9}$ |
| MNIST | EVEN | ODD | $9.7_{\pm0.6}$ | $19.2^{(71.1)}_{\pm2.3}$ | $11.5^{(71.1)}_{\pm1.2}$ | $10.1^{(70.0)}_{\pm0.7}$ | $\mathbf{66.8}_{\pm0.8}$ | $67.8_{\pm0.5}$ |
| BEER REVIEW | LOOK | AROMA | $55.5_{\pm1.7}$ | $31.9^{(70.1)}_{\pm1.0}$ | $53.7^{(70.1)}_{\pm1.4}$ | $54.1^{(76.0)}_{\pm2.2}$ | $\mathbf{75.9}_{\pm1.4}$ | $77.3_{\pm1.3}$ |
| BEER REVIEW | LOOK | PALATE | $46.9_{\pm0.3}$ | $22.8^{(70.0)}_{\pm1.9}$ | $49.3^{(73.2)}_{\pm2.1}$ | $52.8^{(73.3)}_{\pm2.9}$ | $\mathbf{73.8}_{\pm0.7}$ | $74.0_{\pm1.2}$ |
| BEER REVIEW | AROMA | LOOK | $63.9_{\pm0.6}$ | $40.1^{(68.6)}_{\pm3.1}$ | $65.2^{(66.4)}_{\pm1.8}$ | $64.0^{(71.5)}_{\pm0.6}$ | $\mathbf{80.9}_{\pm0.5}$ | $80.1_{\pm0.6}$ |
| BEER REVIEW | AROMA | PALATE | $46.9_{\pm0.3}$ | $14.0^{(68.3)}_{\pm2.4}$ | $47.9^{(63.2)}_{\pm3.3}$ | $50.0^{(71.2)}_{\pm1.4}$ | $\mathbf{73.5}_{\pm1.1}$ | $74.0_{\pm1.2}$ |
| BEER REVIEW | PALATE | LOOK | $63.9_{\pm0.6}$ | $40.4^{(57.2)}_{\pm2.8}$ | $64.3^{(60.1)}_{\pm2.7}$ | $63.1^{(75.9)}_{\pm1.0}$ | $\mathbf{81.0}_{\pm1.0}$ | $80.1_{\pm0.6}$ |
| BEER REVIEW | PALATE | AROMA | $55.5_{\pm1.7}$ | $23.1^{(59.2)}_{\pm3.3}$ | $54.5^{(58.7)}_{\pm1.2}$ | $56.5^{(73.3)}_{\pm1.3}$ | $\mathbf{76.9}_{\pm1.5}$ | $77.3_{\pm1.3}$ |
| ASK. | PENE | INCI. | $79.3_{\pm1.3}$ | $71.7^{(72.7)}_{\pm0.5}$ | $79.3^{(71.2)}_{\pm0.8}$ | $71.1^{(73.5)}_{\pm1.4}$ | $\mathbf{83.2}_{\pm1.8}$ | $84.8_{\pm1.2}$ |
| ASK. | INCI. | PENE. | $71.6_{\pm1.8}$ | $64.1^{(83.4)}_{\pm1.5}$ | $72.0^{(83.4)}_{\pm3.1}$ | $61.9^{(82.4)}_{\pm0.7}$ | $\mathbf{78.1}_{\pm1.4}$ | $78.3_{\pm0.9}$ |
| BIRD | WATER | SEA. | $81.8_{\pm4.3}$ | $87.8^{(99.5)}_{\pm1.1}$ | $82.0^{(99.5)}_{\pm4.0}$ | $88.0^{(99.5)}_{\pm0.9}$ | $\mathbf{93.1}_{\pm0.4}$ | $93.7_{\pm0.7}$ |
| BIRD | SEA. | WATER | $75.1_{\pm6.3}$ | $94.6^{(93.3)}_{\pm1.6}$ | $78.2^{(93.1)}_{\pm8.1}$ | $93.5^{(92.7)}_{\pm1.9}$ | $\mathbf{99.0}_{\pm0.4}$ | $98.9_{\pm0.5}$ |
| | Average | | 55.2 | 43.7 | 55.8 | 56.4 | **79.3** | 79.6 |

**Transfer methods**   Since the source task contains multiple environments, we can learn a stable model on the source task and transfer it to the target task. We use four algorithms to learn the source task: DANN (Ganin et al., 2016), C-DANN (Li et al., 2018b), MMD (Li et al., 2018a), PI (Bao et al., 2021). We consider three standard methods for transferring the source knowledge:

REUSE: We directly transfer the feature extractor of the source model to the target task. The feature extractor is fixed when learning the target classifier.

FINETUNE: We update the feature extractor when training the target classifier. (Shafahi et al., 2020) has shown that FINETUNE may improve adversarial robustness of the target task.

MULTITASK: We adopt the standard multi-task learning approach (Caruana, 1997) where the source model and the target model share the same feature extractor and are jointly trained together.

**Automatic de-biasing methods**   For the target task, we can also apply de-biasing approaches that do not require environments. We consider the following baselines:

EIIL (Creager et al., 2021): Based on a pre-trained ERM classifier's prediction logits, we infer an environment assignment that maximally violates the invariant learning principle (Arjovsky et al., 2019). We then apply group DRO to minimize the worst-case loss over all inferred environments.

GEORGE (Sohoni et al., 2020): We use the feature representation of a pre-trained ERM classifier to cluster the training data. We then apply group DRO to minimize the worst-case loss over all clusters.

LFF (Nam et al., 2020): We train a biased classifier together with a de-biased classifier. The biased classifier amplifies its bias by minimizing the generalized cross entropy loss. The de-biased classifier then up-weights examples that are mis-labeled by the biased classifier during training.

M-ADA (Qiao et al., 2020): We use a Wasserstein auto-encoder to generate adversarial examples. The de-biased classifier is trained on both the original examples and the generated examples.

DG-MMLD (Matsuura and Harada, 2020): We iteratively divide target examples into latent domains via clustering, and train the domain-invariant feature extractor via adversarial learning.

**ORACLE**   For synthetic environments, we can use the spurious feature to define groups and train an oracle model. For example, in task SEABIRD, this oracle model will minimize the worst-case

Table 3: Worst-group and average-group accuracy on CelebA. The source task is to predict `Eyeglasses` and the target task is to predict `BlondHair`. We use the attribute `Young` to define two environments: $E_1 = \{\text{Young} = 0\}$, $E_2 = \{\text{Young} = 1\}$. Both environments are available in the source task. In the target task, we only have access to $E_1$ during training and validation.. We show the results for the first 3 attributes (alphabetical order). The right-most Average* column is computed based on the performance across all 38 attributes. See Appendix F for full results.

| | METHOD | ArchedEyebrows | | Attractive | | BagsUnderEyes | | Average* | |
|---|---|---|---|---|---|---|---|---|---|
| | | Worst | Avg. | Worst | Avg. | Worst | Avg. | Worst | Avg. |
| | ERM | 75.43 | 88.52 | 75.00 | 88.94 | 70.91 | 87.09 | 61.01 | 85.07 |
| TRANSFER | REUSE$_{\text{PI}}$ | 53.71 | 64.05 | 52.13 | 64.85 | 52.50 | 66.88 | 47.58 | 64.14 |
| | REUSE$_{\text{DANN}}$ | 59.56 | 72.44 | 62.03 | 72.26 | 64.58 | 73.83 | 55.27 | 72.31 |
| | REUSE$_{\text{C-DANN}}$ | 56.02 | 67.07 | 57.78 | 67.90 | 57.50 | 68.33 | 53.22 | 68.56 |
| | REUSE$_{\text{MMD}}$ | 48.91 | 59.80 | 48.46 | 61.51 | 58.74 | 63.11 | 50.61 | 61.27 |
| | FINETUNE$_{\text{PI}}$ | 71.86 | 87.02 | 72.73 | 87.34 | 62.50 | 84.10 | 63.07 | 85.27 |
| | FINETUNE$_{\text{DANN}}$ | 65.38 | 83.89 | 63.35 | 84.98 | 56.86 | 81.34 | 50.60 | 80.49 |
| | FINETUNE$_{\text{C-DANN}}$ | 73.85 | 88.90 | 75.61 | 89.39 | 75.86 | 88.14 | 62.03 | 85.57 |
| | FINETUNE$_{\text{MMD}}$ | 76.07 | 88.80 | 74.33 | 89.74 | 78.57 | 88.61 | 66.80 | 86.81 |
| | MULTITASK | 69.66 | 86.91 | 72.73 | 87.44 | 70.00 | 85.21 | 64.37 | 85.21 |
| AUTO-DEBIAS | EIIL | 64.71 | 85.12 | 64.43 | 85.96 | 66.67 | 83.90 | 57.62 | 83.22 |
| | GEORGE | 74.73 | 87.89 | 73.66 | 87.70 | 77.78 | 87.97 | 63.34 | 85.04 |
| | LFF | 45.41 | 60.23 | 47.67 | 60.16 | 42.59 | 60.72 | 42.52 | 62.04 |
| | M-ADA | 64.61 | 83.33 | 67.33 | 83.59 | 70.34 | 85.34 | 54.55 | 80.77 |
| | DG-MMLD | 69.51 | 87.38 | 68.42 | 87.50 | 63.41 | 84.78 | 55.69 | 83.51 |
| | TOFU | **85.66** | **91.47** | **88.30** | **92.76** | **90.38** | **92.41** | **84.86** | **91.71** |

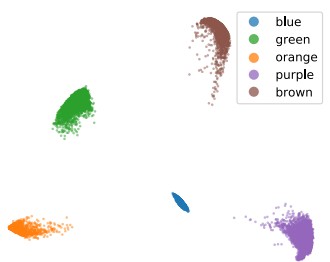

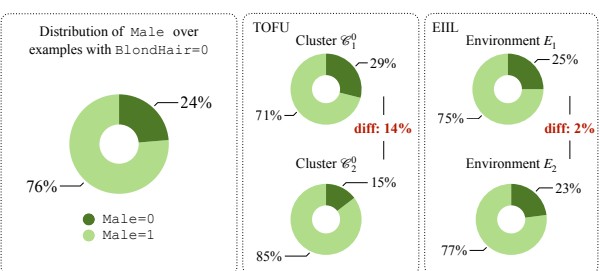

Figure 3: PCA visualization of the unstable feature representation $f_{\mathcal{Z}}$ for examples in MNIST EVEN. $f_{\mathcal{Z}}$ is trained on MNIST ODD. TOFU identifies the hidden spurious color feature by contrasting different source environments.

Figure 4: Visualization of the unknown attribute `Male` on CelebA. Left: distributions of `Male` in the training data. Mid: partitions learned by TOFU. Right: partitions learned by EIIL. TOFU generates partitions that are more informative of the unknown attribute (14% vs. 2%). See Appendix F for results on more attributes.

risk over the following four groups: {seabird in water}, {seabird in land} {landbird in water}, {landbird in land}. This oracle model helps us analyze the performance of our proposed algorithm separately from the inherent limitations (such as model capacity and data size).

## 5 RESULTS

Table 2 summarizes our results on synthetic environments. We observe that standard transfer methods fail to improve over the ERM baseline. On the other hand, TOFU consistently achieves the best

Table 4: Quantitative evaluation of the generated clusters against the ground truth unstable features. For comparison, the TRIPLET baseline directly encourages source examples with the same label to stay close to each other in the feature space, from which we generate the clusters. For both methods, we generate two clusters for each target label value and report the average performance across all label values. We observe that the TRIPLET representation, while biased by the spurious correlations, fails to recover the ground truth unstable features for some tasks. By explicitly contrasting the source environments, TOFU derives clusters that are highly-informative of the unstable features.

| | SOURCE | TARGET | Homogeneity | | Completeness | | V measure | |
|---|---|---|---|---|---|---|---|---|
| | | | TRIPLET | TOFU | TRIPLET | TOFU | TRIPLET | TOFU |
| MNIST | ODD | EVEN | 0.4286 | **0.6820** | 0.5811 | **0.9561** | 0.4906 | **0.7960** |
| | EVEN | ODD | 0.6709 | **0.6764** | 0.9362 | **0.9988** | 0.7814 | **0.8055** |
| BEER REVIEW | LOOK | AROMA | 0.3321 | **0.9256** | 0.2801 | **0.9218** | 0.3039 | **0.9236** |
| | LOOK | PALATE | 0.3317 | **0.9095** | 0.2796 | **0.8988** | 0.3034 | **0.9039** |
| | AROMA | LOOK | 0.3330 | **1.0000** | 0.2811 | **1.0000** | 0.3049 | **1.0000** |
| | AROMA | PALATE | 0.8252 | **1.0000** | 0.7740 | **1.0000** | 0.7974 | **1.0000** |
| | PALATE | LOOK | 0.8306 | **0.9816** | 0.7794 | **0.9811** | 0.8028 | **0.9813** |
| | PALATE | AROMA | 0.8272 | **0.9585** | 0.7764 | **0.9561** | 0.7996 | **0.9573** |

performance across 12 transfer settings, outperforming the best baseline by 22.9%. While TOFU doesn't have access to the unstable features, by inferring them from the source environments, it matches the oracle performance with only 0.30% absolute difference.

Table 3 presents our results on natural environments. This task is very challenging as there are multiple latent spurious attributes in the training data. We observe that most automatic de-biasing methods underperform the ERM baseline. With the help of the source task, FINETUNE and MULTITASK achieve slightly better performance than ERM. TOFU continues to shine in this real-world setting: achieving the best worst-group and average-group performance.

**Is TOFU *able to identify the unstable features?*** Yes. For synthetic environments, we visualize the unstable feature representation produced by $f_{\mathcal{Z}}$ on MNIST EVEN. Figure 3 demonstrates that while $f_{\mathcal{Z}}$ only sees source examples (ODD) during training, it can distribute target examples based on their unstable color features. For natural environments, we visualize the distribution of two latent attributes (Male and ArchedEyebrows) over the generated clusters. Figure 4 shows that the distribution gap of the unknown attribute Male across the generated partitions is 2% for EIIL, only marginally better than random splitting (0%). By leveraging information from the source task, TOFU learns partitions that are more informative of the unknown attribute (14%).

**How do the generated clusters compare to the oracle groups?** We quantitatively evaluate the generated clusters based on three metrics: *homogeneity* (each cluster contain only examples with the same unstable feature value), *completeness* (examples with the same unstable feature value belong to the same cluster), and *V-measure* (the harmonic mean of homogeneity and completeness). From Table 4, we see that TOFU is able to derive clusters that resemble the oracle groups on BEER REVIEW. In MNIST, since we generate two clusters for each label value and there are five different colors, it is impossible to recover the oracle groups. However, TOFU still achieves almost perfect completeness.

## 6 CONCLUSION

Reducing model bias is a critical problem for many machine learning applications in the real world. In this paper, we recognize that bias is a human-defined concept. Without additional knowledge, automatic de-biasing methods cannot effectively distinguish causal features from spurious features. The main departure of this paper is to identify bias by using related tasks. We demonstrate that when the source task and target task share the same set of biases, we can effectively transfer this knowledge and improve the robustness of the target model without additional human intervention. Compared with 15 baselines across 5 datasets, our approach consistently delivers significant performance gain. Quantitative and qualitative analyses confirm that our method is able to identify the hidden biases. Due to space limitations, we leave further discussions to Appendix A.

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
