# OpenReview forum: "Learning Stable Classifiers by Transferring Unstable Features"
_ICLR.cc/2022/Conference — ICLR 2022 Submitted_

### Official Review · Reviewer_tPxR · 2021-10-29

**Correctness:** 4
**Technical Novelty And Significance:** 2
**Empirical Novelty And Significance:** 3
**Recommendation:** 6
**Confidence:** 5

**Main Review:**

Overall I like this paper. This paper does a good job of explaining the setup and the writings are mostly clear. This paper builds upon the core idea from Bao et al. 2021 that uses the error of the model in different environments to capture the spurious patterns, and it further extends to the transfer learning setup, which I believe it's a new setup.

My concerns are as followed:
1. The assumptions are too strong in my opinions, and I think a discussion section will help readers understand the limitations:
 - A) I am skeptical about the assumption in Bao et al. 2021 that uses the error of the model across environments to capture the spuriousness. There could be many factors behind errors across environments such as mislabeled data, different viewpoints of the image (e.g. rotations), noise, and data distribution shifts. None of it corresponds to the unstable features. If this assumption breaks, then there is no real spurious pattern to be learned here.
- B) On the data side, this work assumes the source tasks need to have varying degrees of spuriousness and the target task has the same spurious patterns. How do we possibly check these assumptions are real in the real-world setting?

2. Regarding the demonstrated results
- A) There could be different spurious factors across different classes. And the binarization procedure of correct or incorrect predictions could lead to underspecification since there might be multiple spurious factors but we only treat them as two. Will this method break if there are multiple spurious patterns?
- B) The multi-stage approach of training could be brittle, and selecting hyperparameters could be difficult since no spurious correlation is known beforehand. Maybe the authors can show if this approach is robust when the assumptions are mildly violated like there is a distribution shift among environments, or when there are multiple spurious patterns in the data, and illustrate how to select hyperparameters.

**Summary Of The Paper:**

This paper aims to transfer the knowledge of spurious correlations in a set of source environments to a target environment. They assume the degree of spurious patterns vary among source environments (e.g. color has different correlations with the number in MNIST), and the spurious pattern are the same in both the source and target tasks. Then they aim to transfer the spurious knowledge from source environments to learn a classifier in the target task that ignores the spurious pattern.

They train the model in 3 stages. First, they train a regular classifier in each environment. E.g. given two source environments E1 and E2, they train the corresponding classifiers f1, f2. Second, they use the error of the f1 on E2 to "separate" the E2 examples into two groups, one group for correct predictions and the other group for the wrong predictions under each class. They assume the cause of the error comes from the spurious patterns, so each group will correspond to either high or low spurious patterns learned in f1. Thus they learn a f_Z that outputs an embedding to separate these two groups by triplet loss. Finally, in the target task, they cluster the examples into different groups by the similarity of f_Z that captures examples with similar spurious patterns. Then they apply Distribution Robust Optimization (DRO) on these groups that optimizes the worst performance across all groups. This ignores the spurious patterns based on f_Z in the target task and learns a stable classifier.

**Summary Of The Review:**

Pros:
+ A new setup that transfers the knowledge of spurious patterns to a target task.
+ The writing is clear. Figures and tables are beautifully produced.
+ The results are demonstrated in multiple datasets and the number is convincing compared to several recent baselines. The ablation study of group numbers is great.

Cons:
- The hypotheses in my opinion are a bit too strong. It's unclear in what real-world settings this method will work unless in the contrived setup in this paper. Including a limitation in the paper can strengthen the paper.
- The multi-stage approach of training could be brittle.

I am leaning towards the borderline with marginal acceptance. The experiments are complete with multiple baselines, clear writing and a good ablation study. I feel the assumption is too strong which would need some justifications. And if the authors can show a robust study under different mildly violated assumptions could further strengthen this paper.

---

> ### Author Response · Authors · 2021-11-19
> **Reply to tPxR**
>
> Thank you for your detailed comments. Your suggestions are very helpful!
>
> We have added an additional set of experiments on **real-world environments** where there are **multiple latent spurious features** (see **Update 1** in general reply). We also elaborate the contributions and limitations of our work in Appendix A.
>
> Regarding your concerns:
>
> + 1A. We note that we do not consider data noise and domain gap in Theorem 1 for ease of analysis. However, **these assumptions are relaxed in our empirical experiments**. For example, we explicitly added label noise into the MNIST data. In CelebA, there is a distribution gap (from young people to the elderly) across the two environments. We observe that our method is still able to **perform robustly in situations where the assumption breaks.**
> + 1B. In the real-world setting, we **do not know** what are the spurious features a priori and whether they have varying degrees of spuriousness. For example, in CelebA, we **only know** that the causal relation between the input image and the source task label (Eyeglasses) should be the same for both young people and the elderly. And **the unknown biases that impact generalization from the young people to the elderly should be regularized for the target task** (BlondHair).
> + 2A. Our new results on CelebA demonstrate that our approach is able to handle **multiple spurious factors in real-world environments with distribution shifts**. Compared with 15 baselines, our approach delivers significant performance gain on the worst-case performance (18% across 38 bias attributes).
> + 2B. In terms of hyper-parameter selection, we cluster the validation data based on the learned unstable feature representation $f_z$ and use the worst-group loss as our selection criteria.  Empirically, we observe that our multi-stage approach is very robust during training (Figure 6 presents our model performance across the hyper-parameter search space). Moreover, the multi-stage approach allows us to **easily visualize and understand the bias** (Figure 8), and we can **efficiently adapt** the unstable feature representation to a new target task (since the source task learning is separated from the target task learning).

---

> > ### Comment · Reviewer_tPxR · 2021-11-22
> > **Thank you for the new experiment**
> >
> > Thank you for adding the Celeb-A experiment which strengthens the paper. I also like the discussion, although they are put in the appendix now. I will strongly urge the authors to put a sentence in the conclusion that points the readers to the appendix for limitations and discussions.
> >
> > Q: in Fig. 4 you show that TOFU has difference of 14% while EIIL only has 2%. Why 14% is better than 2% and is more informative? I don't understand the logic here. If I understand correctly, the recovered clusters should correspond to one of the latent factors, and I don't know a larger difference better captures the underlying factors or a lower one. The authors should provide some justifications.
> >
> > After reading the rebuttals and other reviewers' comments, I will still vote for acceptance. It seems that other reviewers share the same criticism that this transfer setup is not realistic, but I think the celeb-A somehow helps address this point. But I'm not sure it's appropriate to add a brand new experiment during the rebuttal process which puts a lot of burden on reviewers' time.

---

> > > ### Author Response · Authors · 2021-11-22
> > > **thanks for the quick response**
> > >
> > > Thank you for taking the time to read our revision and reply to us.
> > >
> > > + We have added a sentence at the end of the conclusion to point the readers to the appendix.
> > > + *A larger gap is more informative of the spurious attribute.*
> > > Let’s consider random splitting as a baseline. The difference between the distributions of the spurious attribute (Male) across the splits is 0% since random splitting partitions the data uniformly. Applying group DRO on these partitions will not provide any improvements on robustness, as the distributions of the two partitions are exactly the same. From Fig. 4, we observe that the partitions generated by EIIL are very similar to random splitting. By leveraging information on the source task, TOFU generates partitions that are much more informative of the spurious attributes (14% vs. 2% on Male, 12% vs. 0% on ArchedEyeBrows, 13% vs. 2% on Attractive).
> > > + We really appreciate that you took the time to reconsider our paper. Thank you!

---

> > > > ### Comment · Reviewer_tPxR · 2021-11-22
> > > > **Thanks for another reply**
> > > >
> > > > That makes sense. Thanks!

---

### Official Review · Reviewer_RJhJ · 2021-11-01

**Correctness:** 4
**Technical Novelty And Significance:** 2
**Empirical Novelty And Significance:** 2
**Recommendation:** 3
**Confidence:** 4

**Main Review:**

Strength:
(1) The authors address a critical point that prevents models from generalization, namely spurious correlation.
(2) The whole pipeline is intuitive and easy to follow, and the empirical results are within expectation.

Weakness:
(1) A critical limitation of this work is the strong assumption on the transferability of the unstable feature. IMO, such an assumption is restrictive in most settings. For the intra-task transfer (where the classification tasks are the same between source and target), finding the invariant part via invariance learning methods is more realistic. As for the inter-task transfer (a wilder setting),  it is very rare that the unstable features and the way they correlate the outcome are exactly the same across the tasks.

(2) The related work and baselines are not discussed thoroughly. First of all, more transfer learning methods should be discussed, not just REUSE and FINTUNE. Moreover, several automatic de-biasing methods [1,2,3] proposed recently could also be considered to involve.

[1] Qiao, F., Zhao, L., & Peng, X. (2020). Learning to Learn Single Domain Generalization. In CVPR.
[2] Matsuura, T. and Harada, T., 2020, April. Domain generalization using a mixture of multiple latent domains. AAAI
[3] J. Liu, Z. Hu, P. Cui, B. Li, and Z. Shen, “Heterogeneous risk minimization”, ICML 2021.

**Summary Of The Paper:**

The paper considers a transfer problem when the spurious correlations of source tasks can be applied to the target task. The authors propose to identify the unstable features on source tasks first and then cluster the target data according to these features. Finally, an invariance-based method are incorporated to eliminated the influence of unstable features.

**Summary Of The Review:**

This paper addresses the spurious correlation by transferring knowledge from source tasks. Although intuitions are provided and empirical effectiveness is illustrated accordingly, the method is restrictive due to the strong assumption on the transferability of unstable features. Several important related work and baselines are also missing.

---

> ### Author Response · Authors · 2021-11-19
> **Reply to RJHJ**
>
> Thank you for your detailed comments!
>
> + **Are biases shared across real world tasks?**:
>   + It is true that the unstable features can be different across different tasks. In this paper, we show that for tasks where the biases are shared, we can effectively transfer this knowledge to obtain a more robust model.
>   + We note that **unwanted biases are shared across many real world applications**. In natural language processing, gender bias and racial bias exist across many tasks such as relation extraction [Gaut et al., 2020], semantic role labeling [Jia et al., 2020], abusive language detection [Ji et al., 2018], sentiment analysis [Kiritchenko et al., 2018]. In computer vision, the same geographical bias exists across different object recognition benchmarks such as ImageNet, COCO and OpenImages [de Vries et al., 2019].
>   + To demonstrate the **real-world applicability** of our approach, we included an additional set of experiments on CelebA where there are **multiple latent spurious attributes** (See **Update 1** in general reply). We consider **two different tasks**: predicting Eyeglasses  (source), predicting BlondHair (target) and we use the attribute Young to define our environments. We note that **while the two classification tasks are different, they may share similar biases such as gender, presence of beard, etc.** We evaluate a model's robustness over all unknown attributes (38 in total). Table 3 shows that our approach **delivers significant performance gain (18%) against 5 auto de-biasing baselines and 9 transfer baselines.**
> + **Related work and baselines**
>   + Thank you for pointing out the missing references. We have added all of them into our related work.
>   + In our new experiments (see **Update 2 & Update 3** in general reply), we have also considered **more transfer and auto-debiasing baselines**: M-ADA [Qiao et al., 2020], DG-MMLD [Matsuura and Harda, 2020], DANN [Ganin et al., 2016], C-DANN [Li et al., 2018a], MMD [Li et al., 2018b]. We note that HRM [Liu et al., 2021] learns an invariant feature-selection mask that is applied to all inputs. Their approach is not directly applicable in our settings as we consider latent spurious features and the receptive fields of these latent features differ from inputs to inputs. **We observe that auto de-biasing methods struggle to identify the latent biases.** By transferring the knowledge of biases instead, **our method is much more robust (worst-group acc 84.86% on CelebA).**
>
> ***
>
> Gaut, Andrew, et al. "Towards Understanding Gender Bias in Relation Extraction." Proceedings of the 58th Annual Meeting of the Association for Computational Linguistics. 2020.
>
> Jia, Shengyu, et al. "Mitigating Gender Bias Amplification in Distribution by Posterior Regularization." Proceedings of the 58th Annual Meeting of the Association for Computational Linguistics. 2020.
>
> Park, Ji Ho, Jamin Shin, and Pascale Fung. "Reducing Gender Bias in Abusive Language Detection." Proceedings of the 2018 Conference on Empirical Methods in Natural Language Processing. 2018.
>
> Kiritchenko, Svetlana, and Saif M. Mohammad. "Examining Gender and Race Bias in Two Hundred Sentiment Analysis Systems." NAACL HLT 2018 (2018): 43.
>
> de Vries, Terrance, et al. "Does object recognition work for everyone?." Proceedings of the IEEE/CVF Conference on Computer Vision and Pattern Recognition Workshops. 2019.
>
> Qiao, Fengchun, Long Zhao, and Xi Peng. "Learning to learn single domain generalization." Proceedings of the IEEE/CVF Conference on Computer Vision and Pattern Recognition. 2020.
>
> Matsuura, Toshihiko, and Tatsuya Harada. "Domain generalization using a mixture of multiple latent domains." Proceedings of the AAAI Conference on Artificial Intelligence. Vol. 34. No. 07. 2020.
>
> Ganin, Yaroslav, et al. "Domain-adversarial training of neural networks." The journal of machine learning research 17.1 (2016): 2096-2030.
>
> Li, Ya, et al. "Domain generalization via conditional invariant representations." Proceedings of the AAAI Conference on Artificial Intelligence. Vol. 32. No. 1. 2018a.
>
> Li, Haoliang, et al. "Domain generalization with adversarial feature learning." Proceedings of the IEEE Conference on Computer Vision and Pattern Recognition. 2018b.
>
> Liu, J., Hu, Z., Cui, P., Li, B., & Shen, Z. (2021). Heterogeneous Risk Minimization. ICML2021

---

### Official Review · Reviewer_L5Kn · 2021-11-01

**Correctness:** 3
**Technical Novelty And Significance:** 2
**Empirical Novelty And Significance:** 3
**Recommendation:** 6
**Confidence:** 4

**Main Review:**

In general, the method seems to work and it seems to be justified (both theoretically, albeit in the appendix, and practically). It is really complicated though (involves many steps) and i fail to make a connection between this and say Domain Invariant Representation learning. For example, if i was to solve the problem of domain generalization (the setup seems really similar to me), a reasonable baseline would be to train a source model that learns an invariantt representation between various environments (using DANN, or CMD or MMD or whatever). But this is not in the baselines. Why is that? One difference i "kinda" see is that domains are more or less pure - eg domain is really a different background. The way i read it is that environement is more mixed - it can contain multiple backgrounds. It would be nice to draw comparison between your method and DIRL or at least explain why DIRL is not applicable


Pros:
- The paper is well written and easy to follow (albeit a lot of helpful info is also in the Appendix)
- The experimental results seem convincing, there is a study of dependence of (some) hyperparameters on the end result (num of clusters)

Cons:
- DRO needs at least a brief introduction
- Experiments are only on 2 environements (domains) in train
- The method seems extremely costly: for a number n of source environements, you train n classifiers, then for n^2 pairs you do partions, then you learn fz representation, then cluster the target data all while tuning and looking for fz dimension and number of clusters.
-There are hyperparameters that need to be tuned: fz dimension, number of clusters. Tuning all the hyperparameters using limited target data - how does it work? Do you tune the fz dimension (i could not understand it from text, i think u do tune the num of clusters but not sure about fz dimension)



Additional comments:
- For figure 1, source task environments actually look the same to me tbh: the red is mostly correlated with 0 and green is mostly correlated with 1 for both of the environments. If they were flipped between the environments,i would expect Domain invariant representation learning to be able to filter out the color from the embedding layer
- what if target is also a mix of different environments (you seem to assume that it is all comes from one domain/env
- How practical is experimental setup you are testing on? For your experiments train was coming from 2 environments that were created artificially

Minor: then we uses=>then we use

**Summary Of The Paper:**

The paper is concerned with learning a model in situations when some features have spurious correlations with the label (for example, for classifying sheep vs camel, background can be a spurious feature). The idea is that if a lot of source data is available, these spurious correlation features should be easy to find. To find the unstable features, authors hypothesize that they are related to mistakes that a classifier makes in different environments. Therefore, they learn a model on one environment and run it on a different environment (source data), splitting the data into correctly predicted and not. Then for each partition (correct or not) examples with the same labels are encouraged to be close together, so the embedding learns UNSTABLE features (fz). Then the target data is clustered based on unstable features representation fz, and then DRO (existing method that assumes the existance of correct groupping based on unstable features) is used to train a robust target classifier

**Summary Of The Review:**

Please see main review

---

> ### Author Response · Authors · 2021-11-19
> **Reply to L5Kn**
>
> Thank you for your detailed comments!
>
> + **Practical utility:** We have added a new set of experiments on CelebA where environments are **directly defined** by a given input attribute (Young) (See **Update 1** in general reply). In Table 3, we observe that our approach is able to **reduce biases from 38 unknown spurious attributes, outperforming the best baseline by a large margin (18%).**
> + **Connection to DIRL:** Domain invariant representation learning aims to match the feature across domains. However, the domain-invariant representation can **still encode spurious features.**  For example, in CelebA, the *Gender* attribute is spuriously correlated with the label *BlondHair*. Given the two environments {Young=0} and {Young=1}, DIRL learns an *age-invariant representation*. However, if the distribution of Gender is the same across the two environments, **DIRL will encode this Gender attribute into the age-invariant representation** (since it is helpful for predicting the target BlondHair attribute).
> In our approach, we only assume that the correlations between Gender and BlondHair are different in the two environments (The elderly may have more white hair). **Even though the distribution of Gender may be the same, we can still identify this bias from the classifiers’ mistakes.** We have also **added DANN, C-DANN, MMD** into our transfer baselines (See **Update 3** in general reply). While these methods improve over the ERM baseline, they still perform poorly on minority groups (worst case acc 66.80% on CelebA).
> + **Time complexity:** Our method can be **efficiently generalized** to multiple environments. Specifically, given $N$ source environments, the $N$ environment-specific classifiers can be learned efficiently with multi-task learning (one shared representation backbone with $N$ different classifier heads). This significantly reduces the inference cost from $O(N^2)$ to $O(N)$ as we only need to pass the input through the representation backbone for one time. Given the partitions, the computation cost of the remaining procedures is $O(N)$ as we can sample the partitions during training. In this paper, we focus on the two-environments setup for simplicity and leave this generalization to future work. We have added this discussion to our appendix.
> + **Hyper-parameter tuning:**  We clarified our hyper-parameter tuning procedures in Appendix C and present our hyper-parameter search space in Figure 6. For the experiments of our method in Table 2 and 3, we **only tuned the learning rate** (1e-3, 1e-4), **weight decay** (1e-1, 1e-2, 1e-3 for vision dataset) and **dropout** (0.1, 0.3, 0.5 for nlp dataset). We **do not tune the dimension of f_z** (it has the same architecture as the representation backbone that is shared across all methods). Based on our analysis in Figure 5, we fix the number of clusters to be 2 and do not tune this parameter.
> + **Comment on Figure 1:** Yes, you are right. If the correlations were flipped between the environments, DIRL will do well. However, **this may not be the case in reality**. For example, in CelebA, women are more likely to have blond hair regardless of the environment (Young). As a result, DIRL baselines fail to reduce such biases.
> + **What if the target is also a mix of different environments?**  For the source task, we assume that different environments exhibit different levels of spurious correlations. For the target task, since we are only given one collection of examples, one can view this either as a single environment or a mixture of N different environments. For example, in the WATER task, you can think of $E_1^{train}$ as a single environment where the correlation between the background and label is 0.36. You can also think of it as a mixture of two environments where the spurious correlation is 1 in the first environment and -1 in the second environment.

---

> > ### Comment · Reviewer_L5Kn · 2021-11-29
> > **Thanks for your responses**
> >
> > Dear authors
> > Thank you for your responses
> > - New experiment are interesting and do seem support the practicality
> > - Thank you do DANN and other DIRL experiments. I do think this discussion should be in the main paper though
> > - Time complexity with multitask head should be mentioned
> > I am bumping my score up
> > Thanks

---

### Official Review · Reviewer_J8M5 · 2021-11-02

**Correctness:** 4
**Technical Novelty And Significance:** 2
**Empirical Novelty And Significance:** 2
**Recommendation:** 3
**Confidence:** 4

**Main Review:**

(1) Strength: The problem that this paper addressed is important and essential for machine learning. And the whole pipeline is reasonable and achieves good results.
	(2) Weakness:
		(a) Method: The proposed method infers the unstable features on the source task and then learns the stable correlation on the target leveraging the knowledge of unstable features. As for the methods, it uses the Group-DRO as the backbone and clusters the available testing data to produce the group labels. However, I think the method is quite ad-hoc and naïve, and it lacks technical contributions. The idea of clustering data with unstable features is also quite similar to the HRM[1], which further reduces the contributions.
		(b)Problem Setting: This method requires both labeled data from the source task and target task, which uses much much more information than existing methods for OOD generalization.
		(c)Theoretical Analysis: There is almost no theoretical analysis for the proposed method. Why it can infer the unstable features? Why this method can generalize to unseen domains? What if the worst-case group also reflects some spurious correlations?
		(d)Experiments: Since it is both the OOD generalization method and the transfer learning method (setting is the same), there lack many baselines. I think more transfer learning methods should be taken into consideration, as well as some domain generalization methods.
		(e)Definitions: What do the stable features or unstable features mean in this paper? What is the difference between causal features[2], invariant features[3], or stable features[4][5]? What is the formal definition of stable and unstable? If it is just the same as causal features, why use a totally new name? I think the authors missed many related works here.

[1] Liu, J., Hu, Z., Cui, P., Li, B., & Shen, Z. (2021). Heterogeneous Risk Minimization. ICML2021
[2] Peters, J., Bühlmann, P., & Meinshausen, N. (2016). Causal inference by using invariant prediction: identification and confidence intervals. Journal of the Royal Statistical Society. Series B (Statistical Methodology), 947-1012.
[3 Koyama, M., & Yamaguchi, S. (2020). Out-of-distribution generalization with maximal invariant predictor. arXiv preprint arXiv:2008.01883.]
[4] Kuang, K., Xiong, R., Cui, P., Athey, S. and Li, B., 2020, April. Stable prediction with model misspecification and agnostic distribution shift. AAAI 2020
[5] Shen, Z., Cui, P., Liu, J., Zhang, T., Li, B. and Chen, Z., 2020, August. Stable learning via differentiated variable decorrelation. KDD 2020


**Summary Of The Paper:**

This paper proposes a two-stage method for transfer learning, which firstly infers unstable features from the source task and then learns stable correlations for the target. Experimental results validate the effectiveness of the proposed method.

**Summary Of The Review:**

This paper proposes a two-stage method for OOD generalization and transfer learning problems. However, there exist many weaknesses, including method, problem setting, theoretical analysis and experiments.

---

> ### Author Response · Authors · 2021-11-19
> **Reply to J8M5**
>
> Thank you for your detailed comments!
>
> + A. Method
>   + A.1. We would like to emphasize that this paper makes a unique and valuable contribution to the de-biasing community. The main contribution of this paper is not a specific learning algorithm, but rather the **idea of transferring the knowledge of biases across related tasks**. Theoretically, we showed that by contrasting different source environments, we can infer the unstable features. Empirically, we demonstrate that our method vastly outperforms 15 baselines across 5 datasets.
>   + A.2. In our related work section, we recognized that partitioning the training data has been widely adopted in the de-biasing community [Creager et al. 2021, Sohoni et al. 2020, Ahmed et al. 2020, Matsuura and Harada 2020, Liu et al. 2021]. Different from prior work, we transferred the knowledge of unstable features from the source task and used it to cluster the target training data.
>   + A.3. Thank you for pointing out the missing reference [Liu et al., 2021]. We have added it into our related work.
> + B. Problem Setting
>   + Bias is a **human-defined concept** and may vary from tasks to tasks. As we pointed out in the related work section, without additional knowledge, automatic de-biasing methods **cannot effectively distinguish causal features from spurious features**. In this paper, **we leverage relations across different tasks to define the exact bias that we want to regularize**. On CelebA (see Update 1), our approach delivers a significant performance gain over five recent auto de-biasing methods (21.52% in absolute accuracy across 38 unknown spurious attributes).
> + C. Theoretical Analysis
>   + C.1. Theorem 1 shows that when we apply a classifier trained on one source environment to another source environment, its mistakes are informative of the unstable features. This enables us to organize examples based on their unstable features (Eq 1).
>   + C.2. We do not consider domain gaps for ease of analysis. If the domains of the source and target tasks are very different, we can extend the algorithm by incorporating an additional domain adversary [Ganin et al., 2016]. We have included this extension in our discussion section.
>   + C.3. Since we cluster the target training data separately for each label value (T.1 on Page 5), within each group, there is no notion of correlation.
> + D. Experiments:
>   + Based on the suggestions, we have included 2 additional auto de-biasing methods and 6 additional transfer baselines (See **Update 2** & **Update 3** in the general reply) . **Compared with 15 baselines, our approach achieves the best worst-group accuracy across 38 attributes on CelebA.**
> + E. Definitions:
>   + E.1. Thank you for pointing out the missing references [Kuang et al., 2020, Shen et al., 2020]. We have added them into our related work.
>   + E.2. To clarify, there is no difference between causal features [Peters et al., 2016], invariant features [Koyama and Yamaguchi, 2020] and stable features. We defined stable/unstable features at the beginning of our method section. We note that these terms have been used in many previous works [Woodward  et al., 2005, Arjovsky et al., 2019, Bao et al., 2021]. In this paper, we use these terms to emphasize that our bias is defined by the source environments.
>
>
> ***
>
> Ganin, Yaroslav, et al. "Domain-adversarial training of neural networks." 2016
>
> Creager, Elliot, Jörn-Henrik Jacobsen, and Richard Zemel. "Environment inference for invariant learning." 2021
>
> Sohoni, Nimit S., et al. "No subclass left behind: Fine-grained robustness in coarse-grained classification problems." 2020
>
> Ahmed, Faruk, et al. "Systematic generalisation with group invariant predictions." International Conference on Learning Representations. 2020.
>
> Matsuura, Toshihiko, and Tatsuya Harada. "Domain generalization using a mixture of multiple latent domains." Proceedings of the AAAI Conference on Artificial Intelligence. Vol. 34. No. 07. 2020.
>
> Liu, J., Hu, Z., Cui, P., Li, B., & Shen, Z. (2021). Heterogeneous Risk Minimization. ICML2021
>
> Kuang, K., Xiong, R., Cui, P., Athey, S. and Li, B., 2020, April. Stable prediction with model misspecification and agnostic distribution shift. AAAI 2020
>
> Shen, Z., Cui, P., Liu, J., Zhang, T., Li, B. and Chen, Z., 2020, August. Stable learning via differentiated variable decorrelation. KDD 2020
>
> Arjovsky, Martin, et al. "Invariant risk minimization." 2019
>
> Bao, Yujia, Shiyu Chang, and Regina Barzilay. "Predict then Interpolate: A Simple Algorithm to Learn Stable Classifiers." ICML 2021.
>
> Woodward, James. Making things happen: A theory of causal explanation. 2005.
>
> Peters, J., Bühlmann, P., & Meinshausen, N. Causal inference by using invariant prediction: identification and confidence intervals. 2016
>
> Koyama, M., & Yamaguchi, S. Out-of-distribution generalization with maximal invariant predictor. 2020

---

> > ### Comment · Reviewer_J8M5 · 2021-11-29
> > **Still have concerns on the novelty and contributions**
> >
> > Thanks for the authors' reply. And I still find it obvious that this paper's novelty and contributions cannot meet the bar of ICLR.
> >
> > [1] As for the methods, it uses the Group-DRO as the backbone and clusters the available testing data to produce the group labels. As the authors said, they did not propose new or novel algorithms, but only raised the idea of transferring the knowledge of biases across related tasks. However, such idea has already been proposed in many works. For example, in most of the invariant learning methods, one aims to learn some 'invariant' representations or features for better generalization, which is just transferring the knowledge of biases across related tasks by discarding the unstable features. In stable learning methods, one aims to reweight and balance the data distribution to remove the bad influence of unstable features, which is also a very similar idea. In many domain generalization methods, one aims to discard the background features or the style features for better generalization. Therefore, simply raising this idea is not novel and therefore the novelty of this paper is not enough for ICLR.
> >
> > [2] The assumptions of the available data are quite strict, which cannot be realized in real scenarios.
> >
> > The mentioned invariant learning methods of very similar idea:
> > [1] Arjovsky, Martin, et al. "Invariant risk minimization." arXiv preprint arXiv:1907.02893 (2019).
> > [2] Creager, Elliot, Jörn-Henrik Jacobsen, and Richard Zemel. "Environment inference for invariant learning." 2021
> > [3] Koyama, M., & Yamaguchi, S. (2020). Out-of-distribution generalization with maximal invariant predictor.
> > [4] Liu, J., Hu, Z., Cui, P., Li, B., & Shen, Z. (2021). Heterogeneous Risk Minimization. ICML2021
> >
> > The mentioned stable learning methods of very similar idea:
> > [1] Kuang, K., Xiong, R., Cui, P., Athey, S. and Li, B., 2020, April. Stable prediction with model misspecification and agnostic distribution shift. AAAI 2020
> > [2] Shen, Z., Cui, P., Liu, J., Zhang, T., Li, B. and Chen, Z., 2020, August. Stable learning via differentiated variable decorrelation. KDD 2020
> >
> > The mentioned domain generalization methods of very similar idea:
> > [1] Y. Ganin, E. Ustinova, H. Ajakan, P. Germain, H. Larochelle, F. Laviolette, M. Marchand, and V. Lempitsky, “Domain- adversarial training of neural networks,” The journal of machine learning research, vol. 17, no. 1, pp. 2096–2030, 2016.
> > [2] H. Li, S. J. Pan, S. Wang, and A. C. Kot, “Domain generalization with adversarial feature learning,” in Proceedings of the IEEE Conference on Computer Vision and Pattern Recognition, 2018, pp. 5400–5409.

---

> > > ### Author Response · Authors · 2021-11-29
> > > **We do not agree with your reply**
> > >
> > > Thank you for the reply. First, we would like to clarify our novelty and contributions:
> > >
> > > 1. **Novelty of our method**:
> > >     + *“However, such idea has already been proposed in many works.”*
> > >       + To the best of our knowledge, **no previous work** includes the idea of transferring the knowledge of biases across tasks into the modeling.
> > >       + Previous work study de-biasing for a **single task**. For example, in invariant/stable learning, the papers that you referred in the reply [Arjovsky et al., 2019, Creager  et al., 2021, Koyama et al., 2020, Liu et al., 2021, Kuang et al., 2020, Shen et al., 2020] address bias by either using multiple data environments or looking at a biased reference classifier. Domain-generalization methods [Ganin et al., 2016, Li et al., 2018] focus on learning a domain-agnostic representation.
> > >       + In this paper, we recognize that many **unwanted biases are actually shared across tasks**. This enable us to learn a robust target model without accessing additional environment information.  Compared against previous work and direct transfer methods, our approach delivers **significant performance gain across 5 datasets.**
> > > 2. **Real world applications**
> > >     + *“The assumptions of the available data are quite strict, which cannot be realized in real scenarios.”*
> > >       + In our paper, we provide multiple realistic experiments and settings in which the assumptions are realized in practice.
> > >       + Unwanted biases are shared across many real world applications. In natural language processing, gender bias and racial bias exist across many tasks such as relation extraction [Gaut et al., 2020], semantic role labeling [Jia et al., 2020], abusive language detection [Ji et al., 2018], sentiment analysis [Kiritchenko et al., 2018]. In computer vision, the same geographical bias exists across different object recognition benchmarks such as ImageNet, COCO and OpenImages [de Vries et al., 2019].
> > >       + We have also conducted experiments on CelebA where there are **multiple latent spurious attributes** (See **Update 1** in general reply). We consider two different tasks: predicting Eyeglasses (source), predicting BlondHair (target) and we use the attribute Young to define our environments. We note that while the two classification tasks are different, they may share similar biases such as gender, presence of beard, etc. We evaluate a model's robustness over **all unknown attributes (38 in total)**. Table 3 shows that **our approach delivers significant performance gain (18%) against 5 auto de-biasing baselines and 9 transfer baselines.**
> > >
> > > ***
> > > In addition to the above clarifications, there are a few points in your reply that we **do not agree with**.
> > >
> > > *“As the authors said, they did not propose new or novel algorithms, but only raised the idea of transferring the knowledge of biases across related tasks. However, such an idea has already been proposed in many works.”*
> > >
> > > + To the best of our knowledge, previous invariant/stable learning methods **do not study biases across tasks**. Please point out a specific reference if we missed anything.
> > >
> > > *“In most of the invariant learning methods, one aims to learn some 'invariant' representations or features for better generalization, which is just transferring the knowledge of biases across related tasks by discarding the unstable features”*
> > >
> > > + In the papers that you mentioned [Arjovsky et al., 2019, Creager et al., 2021, Koyama et al., 2020, Liu et al., 2021], they all study biases in a **single task**.
> > >
> > > *“In stable learning methods, one aims to reweight and balance the data distribution to remove the bad influence of unstable features, which is also a very similar idea.”*
> > >
> > > + [Kuang et al., 2020, Shen et al., 2020] assume that **multiple training environments** for the target task are available. In our work, we only assume **input-output pairs** for the target task. We remove the bad influence of unstable features by transferring the information from a related task.
> > >
> > > *“In many domain generalization methods, one aims to discard the background features or the style features for better generalization.”*
> > >
> > > + The papers that you mentioned [Ganin et al., 2016, Li et al., 2018] **do not** aim to discard the background features or the style features. Instead, they try to match the distribution of the features across two domains. The resulting domain-invariant features can **still encode** background/style information.

---

> > > > ### Author Response · Authors · 2021-11-29
> > > > **References**
> > > >
> > > > Gaut, Andrew, et al. "Towards Understanding Gender Bias in Relation Extraction." 2020.
> > > >
> > > > Jia, Shengyu, et al. "Mitigating Gender Bias Amplification in Distribution by Posterior Regularization." 2020.
> > > >
> > > > Park, Ji Ho, Jamin Shin, and Pascale Fung. "Reducing Gender Bias in Abusive Language Detection." 2018.
> > > >
> > > > Kiritchenko, Svetlana, and Saif M. Mohammad. "Examining Gender and Race Bias in Two Hundred Sentiment Analysis Systems." 2018.
> > > >
> > > > de Vries, Terrance, et al. "Does object recognition work for everyone?." 2019.
> > > >
> > > > Arjovsky, Martin, et al. "Invariant risk minimization."(2019).
> > > >
> > > > Creager, Elliot, Jörn-Henrik Jacobsen, and Richard Zemel. "Environment inference for invariant learning." 2021
> > > >
> > > > Koyama, M., & Yamaguchi, S. (2020). Out-of-distribution generalization with maximal invariant predictor.
> > > >
> > > >  Liu, J., Hu, Z., Cui, P., Li, B., & Shen, Z. (2021). Heterogeneous Risk Minimization. ICML2021
> > > >
> > > > Y. Ganin, E. Ustinova, H. Ajakan, P. Germain, H. Larochelle, F. Laviolette, M. Marchand, and V. Lempitsky, “Domain- adversarial training of neural networks,”, 2016.
> > > >
> > > > H. Li, S. J. Pan, S. Wang, and A. C. Kot, “Domain generalization with adversarial feature learning,”, 2018
> > > >
> > > > Bao, Yujia, Shiyu Chang, and Regina Barzilay. "Predict then Interpolate: A Simple Algorithm to Learn Stable Classifiers." ICML 2021.

---

### Official Review · Reviewer_hKdn · 2021-11-02

**Correctness:** 3
**Technical Novelty And Significance:** 2
**Empirical Novelty And Significance:** 3
**Recommendation:** 6
**Confidence:** 4

**Main Review:**

The paper itself is well written, and does a good job at describing the current research landscape of robust feature learning. Some more thought should be given to the example provided of why a method like TOFU is required, since the current one of using class labels to predict color instead of vice versa seems contrived, and undersells the usefulness of the method. The theoretical claims made throughout the paper are sound, but it is not obvious that they should hold in practice. For example, on datasets where there is heterogeneity in terms of the difficulty in predicting samples, i.e. easy, medium, and ambiguous samples, easy and medium samples might end up in X_1 and X_2, and the ambiguous samples in X_3. Thus, the objective in equation (1) could end up optimizing not for the unstable features in X_1 and X_2 to be closer together than those in X_1 and X_3, but rather for easy samples to be closer together than easy and difficult samples in the learned feature space Z. Additionally, it may be possible to improve S.2 by considering not only correctly predicted vs incorrectly predicted, but also the particular errors that were made. Consider a scenario where images of cars are either misclassified as boats or bicycles. Bicycles are more likely to be on the road than boats, so it may be wise to exclude the cars misclassified as bicycles from X_3, since those contain the correct unstable feature being sought, but violate the assumption of theorem 1. It seems like for TOFU to work, the classifiers trained in step S.1 need to be unstable such that they will make sufficient errors on E_2 in order to have sufficient negative samples for the triplet loss objective in S.3. The limitations of metric learning with a triplet loss should be stated since they could have an influence on model selection done throughout the method.

TOFU has fewer requirements in terms of additional annotations compared to other methods, and domain specific prior knowledge is encoded into how the datasets are split into E_1 and E_2. Quantitative analysis of TOFU reveals that it is significantly superior to the many baselines considered, and that is consistent on all tasks and datasets. This is impressive and rare since many methods in the broader deep learning literature improve on some tasks, but do worse on others. It is also satisfying to see how well TOFU performs in terms of clustering metrics compared to representations learned only for the purpose of classifying the source task. Naturally, the latter representations would contain information encoding the spurious features since they are then leveraged to predict the class label, but they seem to be highly ineffective for the purpose of clustering by unstable feature. Having said that, this analysis is a bit confounded by the fact that ERM uses supervised learning whereas TOFU uses metric learning. Additionally, it would be ideal to have a discussion regarding why the performance improvement of TOFU over ERM is not as significant for the Beer Review data compared to MNIST, even though the clustering scores are much better. In general, the difficulty of the tasks considered for the empirical evaluation should be discussed as it seems to vary quite a bit.

The novelty of the work is somewhat limited. S.1 to S.3 is nearly identical to the cited work of Bao et al., except that the objective in S.3 is changed to the triplet loss objective. This objective comes from Theorem 1 which is a novel contribution, but is based on theoretical results also from the cited work of Bao et al. It would also be recommended that more real life examples of the transfer learning scenario in question be provided, since it currently seems as if it was created to show the applicability of the method. Overall, the paper is very strong from an empirical perspective, is well structured and reproducible, but ignores any potential limitations and is limited in its discussion.

Strengths
-Paper is well written and self-contained.
-Impressive empirical results on many tasks/datasets relevant for this application.
-Provides an interesting perspective on combining transfer learning and robust feature learning,

Weaknesses
-Not so clear how the datasets were split to generate different E1 and E2. For example, in the Waterbirds dataset, if the source task is classifying waterfowl, is the only difference between E1 and E2 the percentage of images that have a water background? This is ambiguous in the appendix as well.
-Conclusion is too short, and is missing a discussion of limitations and future directions.
-A clear limitation of the work is that the unstable features in the target task have to be the same ones as in the source task


**Summary Of The Paper:**

The authors introduce a method (TOFU) for learning classifiers that are robust to spurious correlations in a transfer learning setting. They argue that approaches which rely only on (input, label) pairs and use no extra information to identify spurious features could fail to learn a robust model due to insufficient data for the target task. The authors provide scenarios where useful metadata, namely varying environments, could be available for source tasks, ready to be leveraged by the target task classifier for the purpose of identifying a shared bias that might exist across all tasks. To identify unstable/spurious features, the authors follow a 3-step procedure proposed in an existing work. Namely, a classifier is trained on data from environment E1, and is evaluated on data from a second environment E2. The data in E2 is then partitioned according to the correctness of the classifier's predictions on a per-class basis. A metric learning objective using a triplet loss is used to learn a model which embeds samples according to their unstable features with the goal that the unstable features of the correctly predicted samples should be closer together than unstable features between correctly predicted and incorrectly predicted samples. The objective is justified based on prior theoretical work and is illustrated visually. Extensive empirical evaluations are performed, comparing the performance of TOFU against many different baseline methods on both image and text classification tasks.

**Summary Of The Review:**

This work introduces a new scenario where current methods for learning robust features are not capable of leveraging all available data. It also provides a solution in the form of a method called TOFU which has few requirements in terms of additional data annotations compared to existing methods. The authors recognize that bias is a human defined concept, and could vary from dataset to dataset, so it is best to have a method that can identify unstable features in an automated way. TOFU is an extension of an existing work, and has some theoretical motivations, but the underlying assumptions may be too strong. TOFU outperforms all other methods considered by a large margin, and the authors investigate a potential source of its success by confirming that it is indeed able to cluster data well according to unstable features: a requirement for later on doing group DRO. Sufficient details are provided to enable reproducibility, and many baselines are compared against, making the experiments comprehensive. However, the method is not particularly novel, nor is it clear if the scenario presented occurs often in real life. Limitations of the method are not considered, even though there are a few clear ones both technical and empirical.

---

> ### Author Response · Authors · 2021-11-19
> **Reply to hKdn**
>
> Thank you for your detailed comments!
>
> + **Assumptions of the theoretical analysis:** We do not consider data noise and domain gap in Theorem 1 for ease of analysis. However, we note that **these assumptions are relaxed in our empirical experiments**. For example, we explicitly inserted label noise into the MNIST data (Appendix C.1.1). In CelebA, there is a natural domain gap (from young people to the elderly) across the two environments. We observe that **our method is still able to perform robustly in situations where the assumption breaks.**
> + **Triplet loss requires negative examples:** The triplet loss defined in Eq 1 requires negative examples to train. In this work, we assume that the source environments exhibit different levels of spuriousness (therefore there are enough many mistakes). If their distributions are identical (or very similar), then there is little to learn from them. We have clarified this assumption at footnote 4.
> + **Clustering metrics:** Thank you for the suggestions. We have updated the baseline in Table 4 for fair comparison. The new baseline TRIPLET has the same representation backbone as our unstable feature representation, and uses metric learning (Eq 1) to derive its features from the source labels. We observe that while this new baseline improves over the previous ERM baseline, it still underperforms TOFU across all tasks.
> + **Task difficulty:** The representation backbone (a 2-layer CNN) on MNIST is trained from scratch while we use pre-trained representations for other datasets (details are available in the Appendix). Our hypothesis is that the pre-trained representation prevents the ERM model from over-using the biases.
> + **Novelty:** We would like to emphasize that the main contribution of this paper is not a specific learning algorithm, but rather the **idea of transferring the knowledge of biases across related tasks**. Different from the existing work (Section 2), this enables us to pinpoint the exact bias to regularize.
> + **Real-world applicability:** We have also added an additional set of experiments on **real-world environments where there are multiple unknown spurious features**. [See **Update 1** in the general reply]
> + **Data splitting:** On experiments with synthetic environments, we split the data into different environments according to the agreement between the label and the spurious attribute. We have clarified this process in our Appendix and added the script for our data splitting procedure into the supplemental materials.
> + We have added a detailed discussion in Appendix A to elaborate the limitations and future directions of our work.

---

> > ### Comment · Reviewer_hKdn · 2021-11-28
> > **Reply to authors**
> >
> > Thanks to the authors for running new baselines, and performing experiments on an additional dataset. This further confirms the strong empirical results.
> > Here is the degree to which my concerns were addressed:
> >
> > **Triplet loss and sufficient variation between environments**: The authors now state in the revision how important it is for there to be sufficient variation between source environments in order for the triplet loss to have enough negative samples, but this is informal and not actionable. My point about TOFU not being able to differentiate between samples which have unstable features and samples which are more difficult/ambiguous to predict is partially addressed by the new experiment on CelebA. It would be ideal to have theoretical discussion about how ambiguous samples affect the triplet loss since the reason why the model incorrectly predicts them may not be due to unstable features.
> >
> > **Clustering metrics**: The new TRIPLET baseline is convincing in showing that TOFU’s superior clustering performance is due to how the source environments are partitioned, and not simply due to the fact that is uses metric learning in general.
> > Novelty: The novelty of the work remains as a clever application of previous results to the transfer learning scenario considered. The scenario itself still seems slightly contrived.
> >
> > **Task difficulty**: My comments on task difficulty were addressed.
> >
> > The additional experiments while useful only address some questions that were unanswered by the quantitative results in the initial submission, but they do not address the limitations of the method. Thus, I am maintaining my original score, and encourage the authors to further explore the theoretical part and assumptions of TOFU to strengthen the empirical results.

---

### Author Response · Authors · 2021-11-19
**General reply**

Thank you for your detailed comments and helpful suggestions. We would like to summarize the main contributions of our paper, and our recent updates.

***

One of the key challenges in handling biased data is **separating causal features from spurious signals**. Since such a determination cannot be done just analysing the dataset in isolation, current methods often utilize additional sources of information defining the bias (e.g., environments or biased features). The proposed method enables us to **identify the bias without the requirement of these extra sources of annotation**. Instead, we are **learning from other related tasks where the source of bias is determined**. Compared with 15 baselines on 5 datasets, our method consistently delivers significant performance gain. Both our code and data splits are available for reproducibility.

***

We **highlight** our changes in **red** in the updated pdf for clarity. Here is a brief summary of the major updates:

+ **Update 1** Additional experiments on real-world environments where **multiple latent spurious** features are present (Table 3): We study CelebA where each input is annotated with 40 binary attributes. We consider two tasks: predicting Eyeglasses (source) and predicting BlondHair (target). We use the attribute Young to define two environments: E_1:{Young=0}, E_2: {Young=1}. In the source task, both environments are available. In the target task, only E_1 is accessible during training and validation. At test time, we evaluate the robustness of the model against all other latent attributes (such as Male, ArchedEyebrows, etc.). **We observe that our method outperforms all 15 baselines by a significant margin (18.06%) across 38 bias attributes.**
+ **Update 2** More auto-debiasing baselines: We considered 2 additional auto de-biasing methods: M-ADA [Qiao et al., 2020], DG-MMLD [Matsuura and Harda, 2020]. We observe that both methods struggled to identify the hidden biases directly from the target task (similar to the other 3 auto de-biasing baselines). By transferring the knowledge of biases from the source task, our method vastly outperforms these baselines on CelebA.
+ **Update 3** More transfer baselines: We have also considered domain invariant learning in our transfer baselines: DANN [Ganin et al., 2016], C-DANN [Li et al., 2018a], MMD [Li et al., 2018b]. We observe that while transferring the domain invariant feature representation outperforms ERM, these methods are still very biased (worst-group acc 66.80% on CelebA). By transferring the knowledge of biases instead, our method is much more robust (worst-group acc 84.86% on CelebA).
+ **Update 4** We included a detailed discussion section at Appendix A.

Finally, we have updated the code base with the latest experiments.



***

Qiao, Fengchun, Long Zhao, and Xi Peng. "Learning to learn single domain generalization." Proceedings of the IEEE/CVF Conference on Computer Vision and Pattern Recognition. 2020.

Matsuura, Toshihiko, and Tatsuya Harada. "Domain generalization using a mixture of multiple latent domains." Proceedings of the AAAI Conference on Artificial Intelligence. Vol. 34. No. 07. 2020.

Ganin, Yaroslav, et al. "Domain-adversarial training of neural networks." The journal of machine learning research 17.1 (2016): 2096-2030.

Li, Ya, et al. "Domain generalization via conditional invariant representations." Proceedings of the AAAI Conference on Artificial Intelligence. Vol. 32. No. 1. 2018a.

Li, Haoliang, et al. "Domain generalization with adversarial feature learning." Proceedings of the IEEE Conference on Computer Vision and Pattern Recognition. 2018b.

---

### Decision · Program_Chairs · 2022-01-20

**Decision:**

Reject

**Comment:**

The idea of learning unstable features from source tasks to help learn stable features for a target task is interesting and well-motivated. As the proposed method and its theoretical analysis of learning unstable features from tasks are an incremental extension of an existing work [Bao et al. 2021], the technical contributions line in applying the idea of stable and unstable features learning to the setting of transfer learning. Therefore, the evaluation of this work is focused on the effectiveness of the proposed method in the transfer learning setting.

In transfer learning, one major goal is to make use of knowledge extracted from source tasks to help learn a precise target classifier even with a few or no labeled examples of the target task. It would be more convincing if experiments are conducted to show how the performance of the proposed method changes when the size of labeled data of the target task changes. This is to verify whether the exploitation of unstable features can help to learn a stable classifier for the target tasks more efficiently (i.e., with fewer labeled examples). In addition, as some baseline methods used for comparison do not need to access any labeled data of the target task (like unsupervised domain adaptation or domain generalization approaches), it is not fair to conduct comparison experiments in the setting where there are sufficient labeled examples of the target task since the original designs of such baselines may fail to fully exploit label information in the target task.

Another concern is whether the proposed method is realistic for real-world transfer learning problems. Though in the rebuttal, the authors provided experimental results on a natural environment (CelebA), the constructed transfer learning problem is more like a toy problem. Indeed, there are many transfer learning benchmark datasets that contain multiple domains/tasks. It would be more convincing if experiments are conducted on those datasets.

By considering the above two concerns, this paper is on the borderline. My recommendation is a weak rejection based on the current form of this paper. Note that as some references listed by reviewers RJhJ and J8M5 are not really related to the proposed research here, the novelty of the proposed method compared with those references is NOT taken into consideration to make this recommendation.